# Levels of Evidence Supporting United States Guidelines in Pancreatic Adenocarcinoma Treatment

**DOI:** 10.3390/cancers14164062

**Published:** 2022-08-22

**Authors:** Anna Pellat, Isabelle Boutron, Romain Coriat, Philippe Ravaud

**Affiliations:** 1Gastroentérologie et Oncologie Digestive, Hôpital Cochin, AP-HP, 27 Rue du Faubourg Saint Jacques, Université Paris Cité, 75014 Paris, France; 2Centre of Research in Epidemiology and Statistics (CRESS), Inserm U1153, 1 Parvis de Notre Dame, Université Paris Cité, 75004 Paris, France; 3Centre d’Épidémiologie Clinique, Hôpital Hôtel Dieu, AP-HP, 1 Parvis de Notre Dame, Université Paris Cité, 75004 Paris, France

**Keywords:** pancreatic adenocarcinoma, level of evidence, treatment, United States guidelines

## Abstract

**Simple Summary:**

The aim of our work is to describe the level of evidence supporting therapeutic recommendations in United States pancreatic adenocarcinoma guidelines, and its evolution over time. We recorded the level of evidence for each therapeutic recommendation extracted from the American Society of Clinical Oncology and National Comprehensive Cancer Network guidelines. In both United States guidelines, less than 9% of therapeutic recommendations are supported by a high level of evidence. In the National Comprehensive Cancer Network guidelines, there was no significant increase in high level of evidence recommendations over time. However, guidelines authors can only deal with the available evidence to develop recommendations while highlighting the strengths and weaknesses of included studies. There is a need for a more collaborative effort in pancreatic adenocarcinoma treatment to tackle important therapeutic questions and challenge the current framework of evidence.

**Abstract:**

Cancer guidelines are ideally based on high levels of evidence (LOE). We aim to evaluate the LOE supporting recommendations in United States (US) guidelines on pancreatic adenocarcinoma (PDAC) treatment and its evolution over time. We searched for current guidelines from the American Society of Clinical Oncology (ASCO) and National Comprehensive Cancer Network (NCCN) and their prior publicly available versions on societies’ websites and/or MEDLINE. We recorded the LOE and class of recommendation (opinion of the writing panel) for each recommendation. We defined high LOE as: a “high” quality of evidence from the GRADE methodology (ASCO) and “Category 1” (NCCN). Our main outcome was the proportion of PDAC recommendations supported by high LOE. Proportions of high LOE recommendations were 5% (2/40) and 8% (12/153) in current ASCO and NCCN guidelines, respectively. Less than 10% of class I recommendations were based on high LOE. For NCCN guidelines, the proportion of high LOE recommendations did not improve over time and only three recommendations increased their LOE. We identified a small percentage of high LOE recommendations for PDAC treatment in US guidelines. However, guidelines authors can only deal with the available evidence. The current framework of evidence should be challenged with consideration of observational evidence.

## 1. Introduction

Pancreatic cancer, mostly represented by pancreatic adenocarcinoma (PDAC), is the seventh cause of cancer-related deaths worldwide [1,2]. It shows a particularly high toll in Western countries, and is the fourth leading cause of cancer in the United States (US) [3,4]. Its incidence has been rising in the past years, correlating with the increasing age of the population, but still with a lack of knowledge of all risk factors [3,4]. Improvement in diagnosis allowed to classify the disease as potentially curable (localized and borderline disease) or advanced (locally advanced and metastatic disease). Despite the development of various treatments (e.g.*,* surgery, chemotherapy, and radiotherapy), the prognosis of PDAC remains very low with a 5-year survival rate of approximately 10% [1]. New therapeutics are urgently needed.

Comprehensive guidelines are important tools on which any clinician or health care decision maker can rely to make daily treatment decisions [5]. Evidence-based medicine is currently the preferred and most followed approach for the development of clinical guidelines [6]. It stresses the use of evidence based on the results of interventional research, especially randomized controlled trials (RCTs) and meta-analyses of RCTs, where the randomization process enables the comparison of treatments or interventions with the lowest risk of confounding. Therefore, development of guidelines ideally relies on high levels of evidence (LOE) for optimal patient care. Following this paradigm, PDAC guidelines from the American Society of Clinical Oncology (ASCO) and the National Comprehensive Cancer Network (NCCN) integrated an evidence-based system of classification for the development of their recommendations. 

Our aim is to assess the LOE supporting recommendations in ASCO and NCCN guidelines in PDAC treatment, and its evolution over the years.

## 2. Methods

### 2.1. Data Sources and Search Strategy

We performed a systematic review to identify current US guidelines by the ASCO and the NCCN. Current guidelines were identified as the latest available version posted on each society’s website as of 10 October 2021 (Appendix B). 

### 2.2. Eligibility Criteria for Current Guidelines

Only publicly available (open access on society’s website) comprehensive guideline documents were included in this systematic review. Only guideline documents that included recommendations, including key points, organized by a type of classification of evidence clearly highlighted and separated from the rest of the text were included. Current guidelines were also reviewed to identify if there were any references to a previous iteration of the same guideline. We excluded expert consensus documents and provisional clinical opinions. Similarly, focused updates were not included because they were not representative of the evidence for the entire topic.

### 2.3. Search Strategy and Eligibility for Prior Guidelines

We searched for (1) prior guideline documents archived and publicly available on societies’ websites (Appendix B), and if unavailable, (2) we searched MEDLINE through Google Scholar and PUBMED using the following key words: society’s full name, AND “guidelines” AND “cancer of the pancreas” OR “pancreatic cancer” OR “pancreatic adenocarcinoma” OR “pancreatic neoplasms”. We searched for all prior versions of guidelines available for each society, whatever the date of development. We excluded duplicates (for current guidelines), expert consensus documents, provisional clinical opinions, and focused updates. 

### 2.4. Data Extraction and Presentation

Guideline documents were downloaded, and all recommendations with a classification of evidence (level of evidence and class of recommendation) were abstracted by a single reviewer. Abstraction involved simple reporting and did not require judgment on the part of the abstractor. Recommendations were extracted from the main text and/or summary tables and/or algorithms if present. We used a standardized extraction form for extraction (Appendix A). Therapeutic recommendations in each guideline were categorized by disease stage: (1) potentially curable disease (localized and borderline), (2) advanced disease (locally advanced and metastatic) and (3) all stages. Type of treatment was categorized as follows: (1) surgery, (2) chemotherapy, (3) chemoradiotherapy, (4) chemotherapy or chemoradiotherapy, (5) combination of chemotherapy and chemoradiotherapy (6) palliative (including endoscopy, pain medication, etc.) and (7) other (clinical trials, experimental treatments, immunotherapy, etc.). Categorization was conducted by one reviewer and validated with a senior reviewer. Duplicates were excluded, except when they addressed different disease stages or treatment categories. Because the number of recommendations included in each guideline document could differ substantially, we presented proportions of recommendations for each level of classification in each guideline. Finally, to evaluate whether there was a change in the evidence supporting guideline recommendations over time, we compared proportions over time.

### 2.5. Classification of Evidence (Level of Evidence and Class of Recommendation)

The classification of evidence is defined by a level of evidence and a class of recommendation. In the ASCO guidelines, the classification of evidence is defined by the level of evidence based on the GRADE methodology and a strength of recommendations (Appendix A) [7,8,9,10,11,12,13,14,15,16]. In the NCCN guidelines, the classification of evidence is based on the NCCN categories of evidence and levels of consensus (Appendix A).

#### 2.5.1. Definition of High Level of Evidence (LOE)

Since both guidelines did not use the same classification of evidence, we defined high LOE as the following: − “high” quality of evidence from the GRADE methodology used in the ASCO guidelines (confidence that the true effect lies close to that of the estimate), and− “Category 1” from the NCCN guidelines (high-level evidence).

For more details, see Appendix A. All other LOE from each classification were defined as “other” in our work.

#### 2.5.2. Class of Recommendation

Each recommendation was also attributed a class of recommendation which synthetizes the opinion of the guideline writing panel regarding the risks and benefit based on the evidence and other factors. Based on the classification from the American College of Cardiology/American Heart Association Task force [17], we assigned a class of recommendation to each abstracted recommendation: class I recommendations are those for which there is strong evidence, and/or general agreement in favor of an intervention; class II recommendations are those for which there is conflicting evidence or opinion on the effect of an intervention, and class III recommendations are those for which there is evidence and/or general agreement that the intervention is not useful or effective, or that it may be harmful (Appendix A). Attribution was based on the strength of recommendation and level of consensus data for the ASCO and NCCN guidelines, respectively. Attribution was conducted by two reviewers (A.P. and I.B.).

### 2.6. Outcome Measures

#### 2.6.1. Primary Outcome Measure

The proportion of recommendations supported by high LOE in PDAC treatment for each current guideline.

#### 2.6.2. Secondary Outcome Measures

The proportion of current guideline recommendations supported by high LOE for each category of disease stage and treatment.

The proportion of each class of recommendation supported by high LOE in current guidelines.

The evolution of LOE over time for guidelines available with prior versions.

## 3. Results

### 3.1. Characteristics of Current Guidelines

Current ASCO and NCCN guidelines were published in 2016/2020 and 2021, respectively. Of note, the ASCO had published three documents for different disease stages, two in 2016 and one in 2020, that we considered as one comprehensive guideline in our work. A total of 51 and 189 recommendations were identified for ASCO and NCCN current guidelines, respectively (Figure 1). 

Proportions of therapeutic recommendations were 78% (40/51) and 81% (153/189) in the ASCO and NCCN guidelines, respectively (Figure 1 and Table 1). Other recommendations on incidence and epidemiology, diagnosis and staging, and follow-up were excluded.

### 3.2. Outcome Results

#### 3.2.1. Primary Outcome Measure

There were two and twelve therapeutic recommendations based on high LOE in the ASCO and NCCN guidelines, respectively (Table 1). Proportions of therapeutic recommendations supported by high LOE were 5% (2/40) and 8% (12/153) for the ASCO and NCCN guidelines, respectively. 

#### 3.2.2. Secondary Outcome Measures

Regarding disease stage, proportions of recommendations based on high LOE for “potentially curable disease” ranged from 10% (1/10) to 15% (4/27) between both guidelines (Table 1 and Figure 2A,C). Proportions of recommendations based on high LOE for “advanced disease” ranged from 3% (1/29) to 7% (8/119).

Regarding treatment category, proportions of recommendations for “chemotherapy” based on high LOE ranged from 10% (1/10) to 16% (12/76) (Figure 2B,D). Only one other type of treatment was supported by one high LOE recommendation in the ASCO guidelines (PD-1 immune checkpoint inhibitor for dMMR or MSI-H tumors) (Figure 2B,D). 

In the ASCO guidelines, 27 and 13 therapeutic recommendations were classified as class I and II, respectively. A total of 7% (2/27) of class I therapeutic recommendations was based on high LOE. In the NCCN guidelines, 127 and 26 therapeutic recommendations were classified as class I and II, respectively. A total of 9% (12/127) of class I therapeutic recommendations was based on high LOE. There were no class III recommendations in neither ASCO nor NCCN guidelines.

Only NCCN guidelines had prior publicly available versions of the whole comprehensive guideline, published in 2010 and 2017. There were 59 and 102 therapeutic recommendations in the 2010 and 2017 versions, respectively. The proportion of therapeutic recommendations based on high LOE went from 5% (3/59) in 2010 to 10% (10/102) in 2017 to 8% (12/153) in 2021.

Regarding the evolution of evidence at the recommendation level, only three recommendations evolved from a lower LOE to a high LOE between the 2010 and 2017 guidelines, and none between the 2017 and 2021 guidelines (Figure 3). Between 2017 and 2021, no recommendation evolved from a lower LOE to a high LOE. Other high LOE recommendations came from new evidence for new treatments over time. 

## 4. Discussion

In the two current US guidelines on PDAC treatment, the proportion of recommendations supported by high LOE is small, and the distribution of high LOE recommendations is not homogeneous across different disease stages and types of treatment. When comparing prior and current NCCN guidelines, the proportion of therapeutic recommendations with high LOE did not meaningfully improve over time.

Our work is the first to study the LOE supporting US therapeutic guidelines in PDAC management. In 2016, a panel of expert pancreatologists found that, out of thirty-six clinical questions on pancreatic cancer management, only four had sufficient evidence in available guidelines for agreement [18]. Other works on pancreatic cancer international guidelines mainly focus on their methodological quality using The Appraisal of Guidelines for Research and Evaluation (AGREE II) tool [19,20]. For both diagnosis and therapeutic guidelines, these works reported a suboptimal methodological quality across different guidelines with important variations in recommendations [19,20]. 

We also found great heterogeneity for evidence between PDAC categories. Indeed, most recommendations with high LOE were found for “potentially curable disease” and “chemotherapy”. These results underline the lack of evidence in PDAC for some types of interventions or some stages of the disease where RCTs are probably harder to conduct. Regarding classes of recommendation, only 7% (2/27) and 9% (12/127) of class I recommendations were based on high LOE. In other words, most of strongly recommended interventions are not currently supported by high LOE. 

When comparing prior and current versions of NCCN guidelines, there was no meaningful improvement of the proportion of high LOE recommendations between 2010 and 2021. One recent study focusing on the evolution of evidence underlying NCCN recommendations for a selection of solid cancers between 2010 and 2019 also reported an absence of change in proportions of Category 1 recommendations over time [21]. This is in line with our findings. Furthermore, our work shows that most PDAC recommendations initially based on lower LOE did not increase to a high LOE over time. Indeed, only three recommendations in 2010 evolved to a high LOE in 2017 and none between 2017 and 2021; other high LOE recommendations were new additions. This suggests that there has not been new strong evidence for routinely used treatments in PDAC since 2010, and that most trials probably focus on new therapeutic options. 

Our work has several limits. First, there is heterogeneity between the two current guidelines regarding the total number of therapeutic recommendations. Indeed, in the NCCN guidelines, “resectable” and “borderline” diseases are separately described with dedicated recommendations. Additionally, the NCCN guidelines are constructed as a set of available options that are intended to help the physician choose the optimal treatment in a personalized manner for each patient. These reasons probably explain why there is a higher number of recommendations in the NCCN guidelines. Second, both guidelines use two different classifications of evidence based on different development methodologies. The definition of “high-level evidence” used by the NCCN for their Category 1 is not explicit in their guideline development manual. Therefore, even if our work was purely descriptive, we had to assume that what authors considered as high-level evidence was comparable for both guidelines. Third, the evidence supporting major society guideline recommendations is a surrogate for the totality of the evidence rather than a direct measurement with an inherent risk of subjectivity from each panel of experts involved in their development. 

Our results show a need for improvement in the evidence supporting PDAC therapeutic guidelines, not only for future treatments, but also for the ones that are currently used. Indeed, guidelines authors can only deal with the available evidence to develop recommendations, while highlighting the strengths and weaknesses of included studies. Previous works already pointed out the flaws in the current research system, including fragmentation with a lack of common goal or collaboration, heterogeneity of financial investment, and more reliance on industry funding, often resulting in narrowly focused trials with highly selected patients designed for drug approval, rather than to provide evidence for patients and decision makers [22,23,24,25]. Moreover, the lack of feedback between primary research producers and systematic reviewers precludes the optimal use of the data [24]. First, as discussed in other fields [26,27], when RCTs are inadequate or hard to conduct, we could rely on high quality observational studies by using large registries or routinely collected data, or even conduct RCTs within registries [28,29,30]. One might also argue that high LOE is not essential for all therapeutic recommendations included in clinical guidelines. Therefore, collaborative group efforts should be encouraged to help identify the important clinical questions where strong evidence is currently lacking and focus future trials on these issues. Second, the methodology for development of classifications of evidence might need reconsideration for more transparency and homogeneity. Finally, our work could also be done for European guidelines and national guidelines from various countries for comparison.

## 5. Conclusions

In conclusion, among current PDAC therapeutic recommendations in major US society guidelines, only a small percentage are supported by high LOE. Comparison with prior versions of comprehensive NCCN guidelines showed no meaningful increase of evidence over time. We need more collaborative effort to identify important clinical questions, but also to challenge the current framework of evidence in the context of an increasing number of diverse sources of data.

## Figures and Tables

**Figure 1 cancers-14-04062-f001:**
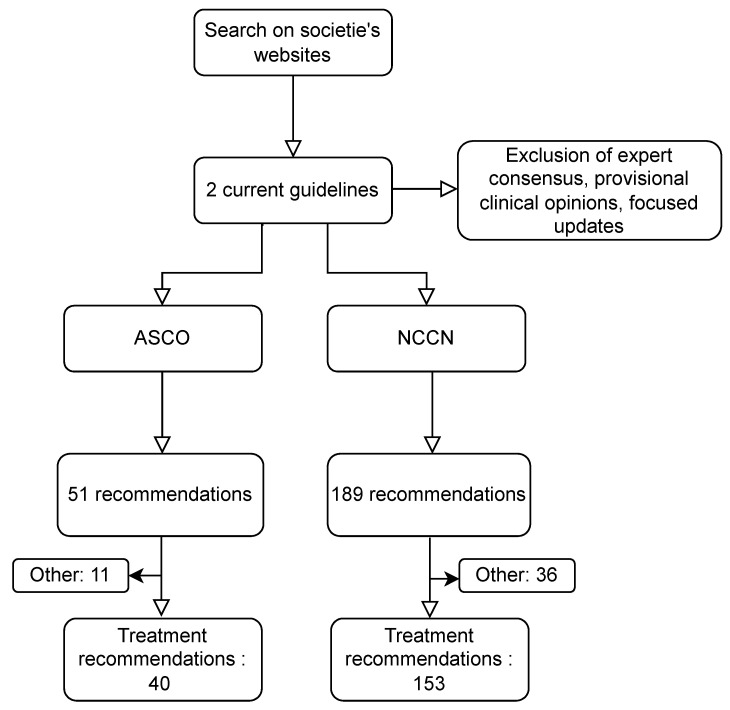
Flow chart of the search for current guidelines and their therapeutic recommendations. ASCO: American society of clinical oncology, NCCN: national comprehensive cancer network. “Other” includes non-therapeutic recommendations on incidence and epidemiology, diagnosis and staging, and follow-up.

**Figure 2 cancers-14-04062-f002:**
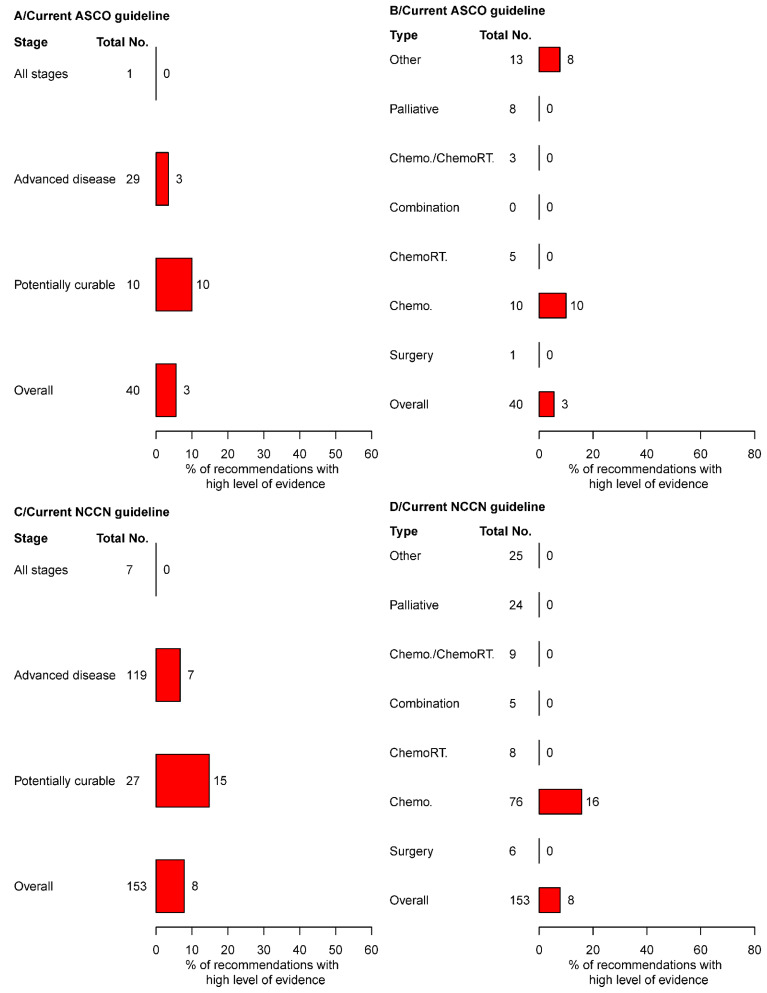
Proportion of therapeutic recommendations classified by high level of evidence in current ASCO and NCCN guidelines: overall, per disease stage and per treatment category: (**A**,**B**) results for the ASCO guidelines; (**C**,**D**) results for the NCCN guidelines. Total No. represents the number of treatment recommendations per disease category. No.: number, ASCO: American society of clinical oncology, NCCN: national comprehensive cancer network.

**Figure 3 cancers-14-04062-f003:**
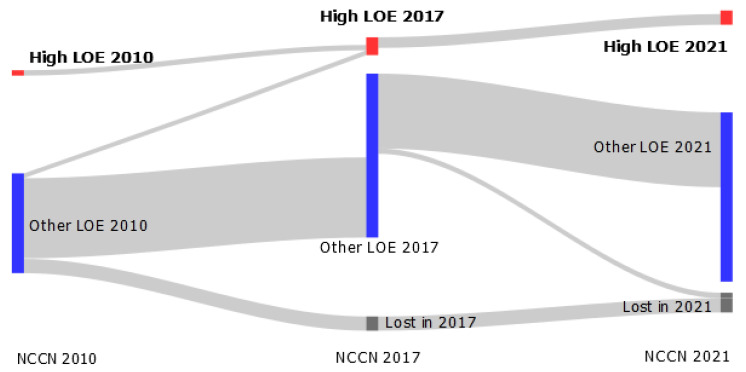
Evolution of evidence at the recommendation level in current and prior NCCN guidelines. LOE: level of evidence. “Lost in”: some recommendations were not repeated in newest versions of NCCN guidelines; “Other LOE” represents all other levels of evidence (see Section 2).

**Table 1 cancers-14-04062-t001:** Number of therapeutic recommendations in current pancreatic adenocarcinoma guidelines, overall and by categories (disease stage and type of treatment).

	ASCO			NCCN		
Level of Evidence	Total	High	Other	Total	High	Other
**Therapeutic** **recommendations**	40	2	38	153	12	141
**Disease stage**						
Potentially curable	10	1	9	27	4	23
Advanced	29	1	28	119	8	111
All stages	1	0	1	7	0	7
**Treatment category**						
Surgery	1	0	1	6	0	6
Chemotherapy	10	1	9	76	12	64
Chemoradiotherapy	5	0	5	8	0	8
Combination	0	0	0	5	0	5
Chemotherapy or chemoradiotherapy	3	0	3	9	0	9
Palliative	8	0	8	24	0	24
Other	13	1	12	25	0	25

For each guideline, recommendations are classified as high level of evidence or other. ASCO: American society of clinical oncology, NCCN: national comprehensive cancer network.

## Data Availability

The data presented in this study are available in this article.

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
