# Peer review of "Levels of Evidence Supporting United States Guidelines in Pancreatic Adenocarcinoma Treatment"

_cancers, 2022, doi:10.3390/cancers14164062_

Round 1

Reviewer 1 Report

The response of the authors of the manuscript did not completely dispel my doubts about the validity of this type of scientific work, i.e. a simple statistical analysis of a complex problem such as diagnostic and therapeutic guidelines. However, perhaps a wide range of researchers and physicians should be allowed to evaluate this work, and the future citation rate for this paper should give an answer to the quality of research. That is why I currently recommend this manuscript for publication after minor changes.

The authors should include their statement from the discussion also in the abstract, for example:

"We identified a small percentage of high LOE recommendations in PDAC treatment in US guidelines, however, guidelines authors can only deal with the available evidence to develop recommendations, while highlighting the strengths and weaknesses of included studies. There is a need for more collaborative effort in pancreatic adenocarcinoma treatment to tackle important therapeutic questions and challenge the current framework of evidence."

Author Response

Response:
Thank you for this suggestion and for your trust.

We have modified both the simple summary and the abstract, taking the maximum number of authorized words into account:

“Simple Summary: The aim of our work was to describe the level of evidence supporting therapeutic recommendations in United States pancreatic adenocarcinoma guidelines, and its evolution overtime. We recorded the level of evidence for each therapeutic recommendation extracted from the American Society of Clinical Oncology and National Comprehensive Cancer Network guidelines. In both United States guidelines, less than 8% of therapeutic recommendations are supported by high level of evidence. In the National Comprehensive Cancer Network guidelines, there was no significant increase of high level of evidence recommendations overtime. However, guidelines authors can only deal with the available evidence to develop recommendations while highlighting the strengths and weaknesses of included studies. There is a need for more collaborative effort in pancreatic adenocarcinoma treatment to tackle important therapeutic questions and challenge the current framework of evidence. “

Abstract: Cancer guidelines are ideally based on high level of evidence (LOE). We aimed to evaluate the LOE supporting recommendations in United States (US) guidelines on pancreatic adenocarcinoma (PDAC) treatment and its evolution overtime. We searched for current guidelines from the American Society of Clinical Oncology (ASCO) and National Comprehensive Cancer Network (NCCN) and their prior publicly available versions on societies’ websites and/or MEDLINE. We recorded the LOE and class of recommendation (opinion of the writing panel) for each recommendation. We defined as high LOE: “high” quality of evidence from the GRADE methodology (ASCO) and “Category 1” (NCCN). Our main outcome was the proportion of PDAC recommendations supported by high LOE. Proportions of high LOE recommendations were 5% (2/40) and 8% (12/153) in current ASCO and NCCN guidelines, respectively. Most high LOE recommendations were written for “chemotherapy” and “potentially curable disease”. Less than 10% of class I recommendations were based on high LOE. For NCCN guidelines, the proportion of high LOE recommendations did not improve overtime and only three recommendations increased their LOE. We identified a small percentage of high LOE recommendations for PDAC treatment in US guidelines. However, guidelines authors can only deal with the available evidence. The current framework of evidence should be challenged with consideration of observational evidence.

Reviewer 2 Report

The manuscript is properly revised.

Author Response

Response

Thank you

This manuscript is a resubmission of an earlier submission. The following is a list of the peer review reports and author responses from that submission.

Round 1

Reviewer 1 Report

The authors of the manuscript themselves emphasized several limitations of the work in the discussion, and I quote: "There was a heterogeneity between the two current guidelines regarding the total number of treatment recommendations, the choice of evidence classification, disease definition, stage, and year of development." Thus,the zero-one score used by the authors of the manuscript (i.e. high level of evidence versus no high level of evidence) in my opinion is factually incorrect from medical point of view.

Moreover, in relation to the ASCO guidelines, the authors of the manuscript assessed their value from 2016. They did not include the 2020 update on metastatic pancreatic cancer. The NCCN guidelines are rather a set of available diagnostic and therapeutic options that are intended to help the physician choose the optimal treatment, personalized for a particular patient. 

The results of previous clinical trials are as published. Therefore, the work of the guideline authors should be appreciated that they have produced a guidelines that highlights the strengths and weaknesses of these studies, thus facilitating the physician's work on the benefit of the patient.

In my opinion the scientific value of the manuscript is low, no scientific soundness, and the results are not encouraging to merit further research of this kind.

Reviewer 2 Report

The authors evaluated the level of evidence (LOE) supporting therapeutic recommendations in pancreatic adenocarcinoma guidelines from the American Society of Clinical Oncology (ASCO) and National Comprehensive Cancer Network (NCCN) and their prior publicly available versions.

My comments are as follows;

1.      The author conclude that further high-quality observation studies, especially RCTs and meta-analyses of RCTs, are needed to increase the proportion of therapeutic recommendations with high LOE in major US society guidelines. The authors are right; however, it would be virtually impossible to conduct high-quality observation studies for the resolution of all clinical questions because of financial aspects and medical ethics. I think high LOE is not essential in all therapeutic recommendations included in clinical guidelines.

2.      The authors should discuss the difference in the number of recommendations and proportions of recommendations supported by high LOE between the current 2016 ASCO and 2021 NCCN guidelines.

3.      Did the authors include the recommendations regarding endoscopic managements for pancreatic cancer in the current study?

4.    Please explain the senior reviewer in detail. Is the senior reviewer an expert of pancreatic cancer?